# Transcriptomic Analysis Reveals the Opposite Regulatory Effects of WRKY and CAMTA Transcription Factors on Total Tannin Production in *Quercus fabri* Fruit

**DOI:** 10.3390/ijms252313103

**Published:** 2024-12-06

**Authors:** Yuqing Cai, Shifa Xiong, Yangdong Wang, Yicun Chen, Liwen Wu

**Affiliations:** 1State Key Laboratory of Tree Genetics and Breeding, Chinese Academy of Forestry, Beijing 100091, China; cirno1999@163.com (Y.C.); xiongshifa111@163.com (S.X.); wangyangdong@caf.ac.cn (Y.W.); yicun_chen@163.com (Y.C.); 2Research Institute of Subtropical Forestry, Chinese Academy of Forestry, Hangzhou 311400, China

**Keywords:** tannins, acorn, transcriptome, transcription factor, WRKY, CAMTA

## Abstract

Tannins are prevalent compounds found in plant fruits, contributing to the bitter taste often associated with these fruits and nuts, thereby influencing their overall taste quality. Numerous studies have been conducted to investigate the regulatory factors involved in tannin synthesis. Among these factors, transcription factors exhibit the most significant capacity to regulate tannin production as they can modulate the expression of several key enzyme genes within the tannin synthesis pathway. In this study, we focused on acorns from *Quercus fabri*, a species abundant in subtropical China. Utilizing transcriptome data from acorns with previously established significant differences in tannin content, we identified novel genes that are capable of regulating tannin synthesis. Specifically, we discovered one transcription factor from the WRKY family and one from the CAMTA family. Promoter response element analysis revealed that the downstream target genes regulated by these two transcription factors are highly similar, and all play crucial roles as enzyme genes in the tannin synthesis pathway. In addition, by detecting the expression levels of two transcription factor genes and target genes, we found that the two transcription factors regulate the target genes in exactly opposite ways. This study not only identifies new transcription factors involved in the regulation of tannin synthesis but also introduces a novel set of molecular biology techniques aimed at effectively modulating tannin content in plant fruits, thereby enhancing fruit quality.

## 1. Introduction

Plant tannins, also known as plant polyphenols, refer to polyphenolic compounds with a molecular weight of 500–3000 Daltons that can precipitate proteins and alkaloids [1]. They are important secondary metabolites in plants and are widely present in the fruits of various plants. The chemical composition of tannins is complex and can be roughly divided into two types. One is condensed tannins, derivatives of flavanols, which are polymerized by C-C bonds between hydroxyflavan-3-ol and hydroxyflavan-3,4-diol [2]. They have a relatively high molecular weight and stable chemical structure but can be condensed into anthocyanins under hot acid action. Therefore, condensed tannins are sometimes referred to as proanthocyanidins [3]. The other type is hydrolyzed tannins, which are formed by the combination of polyols such as glucose with phenolic acids and their derivatives through ester bonds [4]. Their molecular weight is small, their chemical structure is unstable, and they are easily hydrolyzed to form gallic acid and tannic acid [5].

The synthesis pathway of condensed tannins has been validated in various plants, which is formed from shikimic acid through the shikimic acid pathway, phenylpropanoid pathway, and flavonoid pathway [6]; the synthesis pathway of hydrolyzed tannins has not been fully explored, and only a few key enzymes in the synthesis pathway have been identified, including UDP-glycosyltransferase (UGT), serine carboxypeptidase (SCPL), and carboxylesterase (CXE) [7]. When it comes to tannins, people often think of their astringency. The ability of tannins to bind to proteins is called astringency, which refers to the condensation reaction between tannins and proteins through hydrophobic and hydrogen bonds, resulting in a bitter taste when tasted. In addition, when the human body ingests a certain number of tannins, they will crosslink with proteins in the gastrointestinal tract, causing protein metabolism disorders, affecting the digestibility and utilization of proteins, and causing discomfort to the human body [8].

To reduce the discomfort caused by consuming plant fruits, understanding the regulatory mechanism of tannin synthesis in fruits and reducing their tannin content is a relatively effective biological approach. At present, some corresponding studies have explored the regulatory factors of tannin synthesis in fruits, mainly including environmental regulation, hormone regulation, and transcription factor molecular regulation. Among them, transcription factors are relatively efficient in regulating the synthesis of metabolites due to their ability to activate or inhibit the expression of multiple genes in specific metabolic pathways [9]. Currently, dozens of transcription factors have been reported to be involved in tannin synthesis, belonging to protein families such as Homeodomain, MYB, bHLH, WD40, WIP, and MADS [10,11]. Among them, MYB, bHLH, and WD40 are the most extensively studied transcription factors. MYB transcription factors are a large protein family, and different types of MYB transcription factors participate in different metabolic processes in plants, playing important roles in plant growth and development, organ morphogenesis, and secondary metabolism [12]. The MYB transcription factors involved in tannin metabolism regulation were first reported in *Arabidopsis*, and related studies have found that they mainly regulate the expression of *DFR*, *ANS*, and *ANR* genes [13].

So far, numerous MYB transcription factors such as VvMYB5a, VvMYBPA1, and VvMYBPA2 have been discovered in grapes. Among them, VvMYB5a not only regulates the synthesis of tannins and anthocyanins but also regulates the synthesis of secondary metabolites such as flavonol and lignin [14]. VvMYBPA1 and VvMYBPA2 can regulate the synthesis of tannins by activating the promoters of *LAR*, which catalyzes the synthesis of catechins, and *ANR*, which catalyzes the synthesis of epicatechin [15]. Five MYB transcription factors were also discovered from persimmons, among which DkMYB4 mainly promotes the synthesis of tannins by regulating the expression of *DkF3’5’H* and *DkANR* [6]. Akagi et al. found in their study on DkMYB2 that it mainly participates in the metabolic regulation of tannins in persimmon seeds [16]. bHLH transcription factors are also part of a multi-member family and play important roles in plant growth and development, stress response, signal transduction, and secondary metabolism regulation [17]. Previous studies have found that bHLH in grapes can regulate the synthesis of tannins and anthocyanins in the pericarp and seeds, but it must work in synergy with MYB transcription factors to activate the promoters of three key enzyme genes, *CHI*, *UFGT*, and *ANR* [18].

WRKY transcription factors are a unique class of trans-acting factors in plants, comprising one of the largest transcription factor families. In *Arabidopsis*, a total of 74 WRKY members have been identified, while rice contains 109 WRKY members [19]. Currently, genes from this family have been cloned and functionally analyzed across various plant species. Numerous studies indicate that WRKY transcription factors are involved in a wide array of biological processes, including embryogenesis, seed development, leaf senescence, and hormone regulation, which is a crucial component of many signaling networks [20]. CAMTA is a structurally conserved calmodulin-binding transcription factor that is widely present in multicellular eukaryotes [21]. Within the calcium signal transduction pathway, calmodulin serves as a primary calcium sensor, regulating a series of physiological activities in plants by binding to downstream target proteins [21]. In recent years, researchers have conducted functional studies on CAMTA transcription factors in plants, revealing their significant regulatory roles in plant growth and development, defense responses, stress responses, frost resistance, drought resistance, and hormone signaling pathways [22,23,24].

*Quercus fabri* is a deciduous tree species in the Fagaceae family, widely distributed in natural forests in subtropical regions of China [25]. The fruit of *Q*. *fabri* is called acorn, which has high starch content and rich nutrition, and is a traditional natural food [11]. However, the acorn contains a lot of tannins, which makes it bitter in taste. It needs to be processed to remove tannins before it can be consumed by humans. This processing process is cumbersome and requires a lot of time and water resources, greatly limiting the development of the acorn industry. Therefore, exploring the important regulatory factors of tannin synthesis in the acorn can provide new technological means for improving the quality of acorn and promoting the development of the acorn industry. In this study, we compared the transcriptome data of two kinds of acorns with significant differences in tannin content and identified several new transcription factors that can regulate tannin synthesis in plant fruits.

## 2. Results

### 2.1. Total Tannin Content in Acorns at Different Developmental Stages

Testing revealed that the tannin content of acorns collected in Suichang County, Lishui City, Zhejiang Province, was measured at 76.2237 mg/g, 99.6122 mg/g, and 118.6974 mg/g on 23 August, 10 September, and 23 September, respectively. In contrast, acorns from Wanzhou District, Chongqing, exhibited tannin contents of 81.3968 mg/g, 92.5519 mg/g, and 148.5866 mg/g on the same dates (Figure 1). A comparative analysis of the tannin content data from these two locations indicated that the numerical difference was most pronounced and statistically significant during the final measurement period. Therefore, we selected the transcriptome data from both samples during this period for subsequent comparative analysis.

### 2.2. Transcription Factor Family Analysis

An analysis of the transcriptome data from the two samples revealed that the top ten transcription factor families identified were bHLH, NAC, ERF, MYB-related, C2H2, FAR1, WRKY, B3, MYB, and bZIP (Figure 2 and Appendix A). Notably, extensive research has established the regulatory roles of bHLH and MYB in the synthesis of secondary metabolites in fruits, particularly tannins. The presence of these transcription factors in acorns, numbering over 290, suggested their significant involvement in acorn development, with some functions potentially exhibiting redundancy.

### 2.3. Protein Interaction Network Analysis Discovered Two Important Node Transcription Factors, WRKY and CAMTA

Through the analysis of the protein interaction network across the two samples and the integration of gene differential expression results, we identified 42 proteins that exhibited significant expression differences capable of establishing interactive relationships with other proteins (Figure 3). Notably, we identified two transcription factors, LOC115955609 and LOC115995090, which not only demonstrated significant expression differences but also possessed the highest number of interacting factors. Annotation analysis confirmed that LOC115955609 is a member of the WRKY family, while LOC115995090 is classified within the CAMTA family (Appendix A). The protein interaction network indicated that the transcription factors interacting with LOC115955609 (WRKY) include LOC115959757 (ERF), LOC115994479 (Nin-like), and LOC115955389 (ARF). Furthermore, interactions were also observed between LOC115959757 (ERF) and LOC115994479 (Nin-like), as well as between LOC115994479 (Nin-like) and LOC115955389 (ARF). Additionally, we identified that the transcription factors interacting with LOC115995090 (CAMTA) include LOC115968870 and LOC115968957, both of which are transcription factors belonging to the bHLH family. This finding is consistent with previous studies that indicate that bHLH family members can regulate the synthesis of fruit tannins.

### 2.4. Identification of WRKY and CAMTA Transcription Factor Target Genes

Previous studies have demonstrated that WRKY transcription factors can specifically bind to the cis-acting element known as the W-box in the promoter region of target genes, which corresponds to the TTGAC (C/T) nucleotide sequence. This binding can either promote or inhibit transcription, thereby regulating gene expression. Similarly, CAMTA transcription factors interact with the cis-acting element called the CG-box in the promoter region of target genes, specifically binding to the nucleotide sequences (A/C/G)CGCG(T/C/G) or (A/C)CGTGT to modulate transcription and regulate gene expression. By analyzing the promoter elements of key enzyme genes involved in the tannin synthesis pathway, which exhibit significant differences in expression levels between the two samples, we identified that the promoter of the UDP-glycosyltransferase (UGT) gene *LOC115963067* contains a CG-box cis-acting element. Additionally, the promoter of the serine carboxypeptidase (SCPL) gene *LOC115969198* contains one CG-box and one W-box cis-acting element, while the promoter of *LOC115969101* contains two W-box cis-acting elements. The promoter of the carboxylesterase (CXE) gene *LOC115960284* contains one W-box cis-acting element, whereas the promoter of *LOC115956969* contains three CG-box cis-acting elements. The promoter of the phenylalanine ammonia-lyase (PAL) gene *LOC115953247* features one W-box cis-acting element, while the promoter of *LOC115973406* contains two CG-box and one W-box cis-acting elements. The promoter of the chalcone synthase (CHS) gene *LOC115994574* includes one CG-box and one W-box cis-acting element, whereas the promoter of *LOC115976336* contains a single W-box cis-acting element. Furthermore, the promoter of the leucoanthocyanidin reductase (LAR) gene *LOC115991055* also contains a W-box cis-acting element, while the promoter of *LOC115963778* features one CG-box and three W-box cis-acting elements. The promoter of the anthocyanidin reductase (ANR) gene *LOC115957898* contains one CG-box and two W-box cis-acting elements, while the promoter of *LOC115992511* contains one CG-box and one W-box cis-acting element (Table 1 and Appendix A).

Therefore, it is speculated that the target genes of LOC115955609 (WRKY) may include the important enzyme genes *SCPL* (*LOC115969198* and *LOC115969101*) and *CXE* (*LOC115960284*) in the hydrolyzed tannin synthesis pathway, as well as the important enzyme genes *PAL* (*LOC115953247* and *LOC115973406*), *CHS* (*LOC115994574* and *LOC115976336*), *LAR* (*LOC115991055* and *LOC115963778*) and *ANR* (*LOC115957898* and *LOC115992511*) in the condensed tannin synthesis pathway; the target genes of LOC115995090 (CAMTA) may include the important enzyme genes *UGT* (*LOC115963067*), *SCPL* (*LOC115969198*) and *CXE* (*LOC115956969*) in the hydrolyzed tannin synthesis pathway, as well as the important enzyme genes *PAL* (*LOC115973406*), *CHS* (*LOC115994574*), *LAR* (*LOC115963778*) and *ANR* (LOC115957898 and LOC115992511) in the condensed tannin synthesis pathway.

### 2.5. qRT-PCR Validation of Transcription Factor Gene and Target Gene Expression Levels

The transcriptome database of the two samples indicated that the relative expression level of the WRKY transcription factor gene *LOC115955609* in acorns collected from Suichang County, Lishui City, Zhejiang Province, on September 23 was 248.3, while in acorns collected from Wanzhou District, Chongqing City, the level was 949.4. The relative expression levels of the target gene *LOC115969198* were 362.7 and 1529.8, respectively; for *LOC115969101*, the levels were 602.2 and 4296.4; for *LOC115960284*, they were 544.5 and 1790.5; for *LOC115953247*, the levels were 44.7 and 125.8; and for *LOC115973406*, they were 45.7 and 131.3. Additionally, the relative expression levels of *LOC115994574* were 18,952.9 and 25,887.5, while *LOC115976336* showed levels of 17,487.0 and 22,876.7. The expression levels of *LOC115991055* were 185.1 and 584.0, and for *LOC115963778*, the levels were 633.8 and 979.7. Moreover, the relative expression levels of *LOC115957898* were 220.0 and 918.2, and for *LOC115992511*, they were 2207.5 and 3757.4. These data suggested that the regulation of target genes by WRKY transcription factors was positive.

In the two samples, the relative expression levels of the CAMTA transcription factor gene *LOC115995090* were 1977.7 and 790.6, respectively. The relative expression levels of the target gene *LOC115963067* were 7.6 and 453.6, while the relative expression levels of *LOC115969198* were 362.7 and 1529.8. For *LOC115956969*, the expression levels were 69.7 and 134.1, and for *LOC115973406*, they were 45.7 and 131.3. The relative expression levels of *LOC115994574* were 18,952.9 and 25,887.5. The relative expression levels of *LOC115963778* were 633.8 and 979.7, and for *LOC115957898*, they were 220.0 and 918.2. *LOC115992511* exhibited expression levels of 2207.5 and 3757.4. These data indicated that the regulation of target genes by CAMTA transcription factor was negative.

To verify the accurate expression levels of these genes in the transcriptome database, we conducted qRT-PCR experiments. The results indicated that the expression trends of the genes were consistent with the transcriptome data; however, the values exhibited slight deviations (Figure 4).

## 3. Discussion

Tannin content serves as an important indicator of the quality of various fruits and nuts. For instance, when evaluating the quality of sweet persimmons, it is essential to note that the astringency of these fruits is attributed to the soluble tannins they contain. Thus, the tannin content of persimmons is closely related to the overall quality of the fruit [26]. Similarly, in the study of walnut fruit quality, researchers aspire to enhance the taste by reducing the tannin content present in walnuts [27]. Although numerous research reports have addressed the regulatory factors influencing tannin synthesis, there is a notable scarcity of studies focusing on the regulation of the two transcription factors, WRKY and CAMTA.

Existing studies showed that overexpression of the *OsWRKY23* gene in rice can change the expression levels of senescence-related genes, thereby accelerating leaf senescence [28]. Additionally, *AtWRKY44* plays a role in the development of leaf hair morphology and regulates root hair cell differentiation in *Arabidopsis* [29]. Furthermore, WRKY transcription factors participate in the transcriptional regulation processes that enable plants to adapt to various stresses. By functioning as either positive or negative regulators, they can enhance a plant’s defense against biotic and abiotic stresses [30]. For example, the overexpression of the *OsWRKY22* gene in rice significantly improves resistance to *Magnaporthe oryzae* [31], while the overexpression of *OsWRKY89* enhances resistance to the gray rice planthopper [32]. Moreover, transferring the *GhWRKY39-1* gene from cotton into *Nicotiana benthamiana* has been shown to improve salt resistance and oxidative stress tolerance in the plants [33]. Additionally, the overexpression of the soybean *GmWRKY54* gene in *Arabidopsis* enhances the plant’s tolerance to drought and salt stress [34]. Research on CAMTA demonstrates that the CAMTA gene in tomato is crucial for its development and fruit ripening [35]. The expression of 17 auxin-related genes was upregulated in the *camta1* mutant, resulting in a phenotype characterized by hypersensitivity to auxin in the hypocotyl elongation of the plant. Furthermore, researchers have discovered that CAMTA3 is involved in salicylic acid-mediated plant growth, functioning as a negative regulator in the defense response [36].

In this study, we re-analyzed previous transcriptome data and identified one WRKY and one CAMTA transcription factor from the protein interaction network that may play a crucial role in regulating fruit tannin synthesis. We also screened the target genes associated with each transcription factor. Our findings indicate that both WRKY and CAMTA can regulate key enzyme genes, *SCPL* and *CXE*, in the hydrolyzed tannin synthesis pathway, as well as the important enzyme genes *PAL*, *CHS*, *LAR*, and *ANR* in the condensed tannin synthesis pathway. Furthermore, CAMTA also regulates another significant enzyme gene, *UGT*, in the hydrolyzed tannin synthesis pathway. *PAL* and *CHS* are recognized as critical enzyme genes in the early stages of condensed tannin synthesis in plants, while *LAR* and *ANR* are essential in the later stages [37]. These results demonstrate that both transcription factors can comprehensively regulate the tannin synthesis pathway in plant fruits, thus underscoring their important roles.

The regulatory roles of the two transcription factors are fundamentally similar, yet their regulatory mechanisms differ. Gene expression level analyses indicated that the WRKY transcription factor exerts a positive regulatory effect on target genes, whereas the CAMTA transcription factor imposes a negative regulatory effect. This dichotomy in regulation aligns with the existing literature. Generally, the overexpression of WRKY transcription factor genes can enhance the synthesis of metabolites and bolster resistance to both biotic and abiotic stresses by elevating the expression levels of target genes [30]. Notably, the study demonstrated that the overexpression of the tomato *SlWRKY6* gene significantly upregulates the expression of downstream target genes such as *CAT1*, *CAT2*, *CSD1*, and *GSH1*, which in turn improves the plant’s catalase (CAT) enzyme activity and tolerance to heavy metal stress [38]. Furthermore, Chen et al. reported that when tea trees are subjected to anthracnose pathogen attacks, the expression of WRKY proteins is activated, prompting the tea trees to produce substantial amounts of volatile organic compounds in response to the pathogen’s assault [39]. Tannins, as secondary metabolites in plants, play a crucial role in their response to environmental stress by enhancing the antioxidant defense system and improving resistance to such stresses [40]. Additionally, tannins can inhibit the growth of pathogenic microorganisms, thereby contributing to plant protection [41]. The negative regulatory effect of CAMTA transcription factors was first reported in 2009 in a study by Du et al. [42]. Furthermore, in the loss-of-function mutant of *Arabidopsis CAMTA3*, the expression of the disease resistance-related gene *PR1* becomes constitutive, leading to a significant enhancement of plant disease resistance [43].

In addition, this study also identified transcription factors that may interact with the target transcription factors. For instance, WRKY interacts with transcription factors such as ERF, Nin-like, and ARF, while CAMTA interacts with transcription factors from the bHLH family. The interaction between WRKY and ERF has only recently been reported. Wang et al. found that GmWRKY6 can interact with GmERF1 to jointly regulate the transcription of downstream target genes, ultimately affecting the phosphorus utilization efficiency of plants under low phosphorus stress [44]. However, there are no reports on the interaction between WRKY, Nin-like and ARF and the interaction between CAMTA and bHLH. In the next research study, we will use transgenic and mutant creation techniques to explore the molecular mechanism of these transcription factors regulating target genes and verify the actual regulatory effect of these two transcription factors on tannin synthesis in fruits. In addition, we will continue to study other interacting transcription factors, such as ERF, ARF, and bHLH, to find out whether they can also effectively regulate tannin synthesis in fruits. Ultimately, we will use these measured and effective transcription factors to jointly regulate tannin synthesis in fruits, thereby improving fruit quality. In summary, this study analyzed the transcriptome data of two acorn samples with significantly different tannin contents and uncovered previously unreported WRKY and CAMTA transcripts that regulate the tannin synthesis pathway, exhibiting diametrically opposite regulatory effects. The target genes regulated by both transcription factors were explored, and the expression levels of these transcription factors and their target genes were validated. This research study offers a novel technical approach to effectively regulate tannin content in fruits, thereby enhancing fruit quality.

## 4. Materials and Methods

### 4.1. Plant Materials

*Q*. *fabri* acorns were collected from two subtropical provinces in China: Wanzhou District, Chongqing City, and Suichang County, Lishui City, Zhejiang Province. The collection dates for the acorns in these regions were 23 August (immature stage), 10 September (nearly mature stage), and 23 September (mature stage), 2021. Following the harvest, the peel and cupule were removed, and the samples were thoroughly mixed before being stored in an ultra-low-temperature refrigerator at −80 °C for future use.

### 4.2. Determination of Tannin Content

The principle underlying this analysis is that tannins can reduce tungstomolybdic acid, resulting in the formation of a blue compound in an alkaline solution. This compound displays a maximum absorption at 765 nm, with absorbance directly proportional to the tannin content. Quantification is performed by constructing a standard curve. Initially, approximately 0.1 g of acorns was weighed, followed by the addition of 1 mL of distilled water. The mixture was thoroughly homogenized and extracted in a water bath at 80 °C for 30 min. After extraction, the mixture was centrifuged at 10,000 rpm and 25 °C for 10 min, and the supernatant was collected for testing. Prior to measurement, the microplate reader was preheated for over 30 min, and the wavelength was set to 765 nm. Spectrophotometry was employed to determine the absorbance value (A), from which ΔA was calculated as A − A blank. The standard curve was represented by the equation y = 6.4769x + 0.026, with an R^2^ value of 0.9991, where x denoted the standard concentration (mg/mL) and y represented ΔA. The tannin content (mg/g) was calculated using the following formula:tannin content = (ΔA − 0.026) ÷ 6.4769 × V ÷ W = 0.154 × (ΔA − 0.026) ÷ W(1)
where V was the volume of the extraction liquid (1 mL in this experiment) and W was the mass of the sample (g).

### 4.3. Analysis of Transcription Factor Families in Transcriptome Data

This section involved obtaining assembled Unigene information from the *Q*. *fabri* acorns transcriptome database (NCBI Sequence Read Archive, BioProject number PRJNA1046243). The assembled Unigene data were then compared with information from the Plant Transcription Factor Database (PlantTFDB) to predict the transcription factors present and to categorize them into their respective families.

### 4.4. Protein Interaction Network Analysis

The STRING (Search Tool for the Retrieval of Interaction Gene/Proteins) database (https://string-db.org/ (accessed on 20 August 2024)) is currently the most widely utilized and comprehensive tool for protein interaction network analysis, encompassing both reported and predicted protein interaction data. In this study, we performed protein interaction analysis using the STRING database to visually represent the interrelationships among proteins and to identify key nodes within the network. Upon establishing the interrelationships among all target proteins, we visualized them using Cytoscape (v 3.10.1). Additionally, the color and size of the nodes in the figure were adjusted according to the attribute file (*.PPI.attributes.txt) derived from the gene differential expression analysis.

### 4.5. Screening of Target Transcription Factors and Determination of Target Genes

Transcription factors characterized by large node sizes, which indicate a high number of interacting factors within the protein interaction network, were selected as targets for this study. Significant differences in the expression levels of key enzyme genes involved in the tannin synthesis pathway were observed between the two samples analyzed in the transcriptome database. These observations guided the initial selection of target genes. Subsequently, DNA extracted from *Q*. *fabri* leaves was used as a template to amplify target gene promoter fragments. Using the genome of the *Quercus lobata* as a reference, primers were designed for the amplification of the promoter region of the target gene. A promoter fragment of approximately 2 kb was successfully amplified and sequenced. To conduct a cis-acting element analysis on the promoters of these genes, we utilized the PlantCARE database (http://bioinformatics.psb.ugent.be/webtools/plantcare/html/) (accessed on 2 September 2024), which enabled us to identify response elements pertinent to the target transcription factors. Genes that contained these response elements were selected as target genes for further investigation. The primer sequences utilized in this study are provided in Appendix A.

### 4.6. Determination of Relative Expression Levels of Target Transcription Factor Genes and Target Genes via qRT-PCR Analysis

The relative expression levels of target transcription factor genes and associated target genes were derived from the transcriptome database across two samples. To validate these expression levels, RNA was extracted from the samples, and complementary DNA (cDNA) was synthesized using a reverse transcription kit from Applied Biosystems (Shanghai, China). Specific primers targeting the genes of interest were designed for the qRT-PCR experiments. The *Actin* gene of *Q*. *fabri* was utilized as the internal reference gene, and the expression levels of the target genes were quantified using the 2^−ΔΔCt^ method. Three independent biological replicates were performed for the qRT-PCR analysis. The primer sequences are detailed in Appendix A.

### 4.7. Statistical Analysis

All experiments in this study were conducted with three biological replicates. Data analysis was carried out using SPSS statistical software (version 20.0; SPSS Inc., Chicago, IL, USA). The results are presented as the mean ± standard deviation (SD) of the three biological replicates. *p* values of <0.05 and <0.01 were deemed indicative of significant and extremely significant differences, respectively.

## 5. Conclusions

This study conducted new explorations of important regulatory factors for tannin synthesis in fruits, with a view to providing effective biological technology methods for improving the fruit quality of *Q*. *fabri*, which is widely distributed in subtropical areas of China. In this study, we used transcriptome data from previously identified *Q*. *fabri* fruits with significant differences in tannin content to identify novel transcription factors WRKY and CAMTA that regulate tannin synthesis. Both transcription factors play a crucial role in regulating the expression level of structural enzyme genes in the tannin synthesis pathway. However, they regulate target genes in completely opposite ways. Gene expression level analysis showed that WRKY transcription factors exert a positive regulatory effect on target genes, while CAMTA transcription factors exert a negative regulatory effect. Previous studies have shown that WRKY transcription factors usually regulate target genes in a positive direction, but the way CAMTA regulates target genes is less commonly reported. The next step of the research will be to further verify and analyze the regulatory mechanism of the two selected transcription factors on the target genes and to clarify their actual regulatory effects on tannins in fruits through transgenic technology and the creation of mutants.

## Figures and Tables

**Figure 1 ijms-25-13103-f001:**
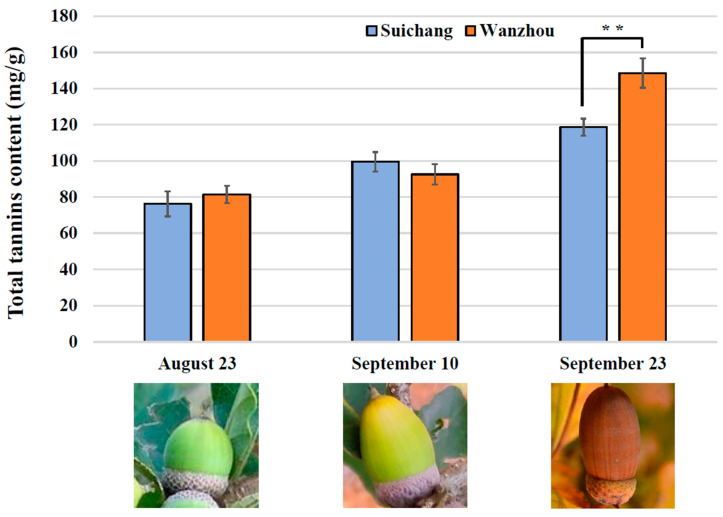
Total tannin content of acorns at different developmental stages in two regions. Data are the mean values. Error bars represent the standard deviation of three biological replicates. Asterisks indicate statistically significant differences based on Student’s *t* test (**, extremely significant difference at *p <* 0.01).

**Figure 2 ijms-25-13103-f002:**
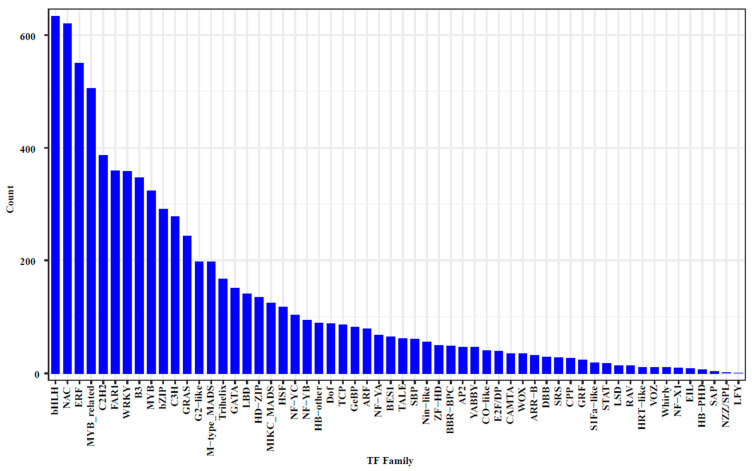
Ranking of transcription factor families based on transcriptome data analysis.

**Figure 3 ijms-25-13103-f003:**
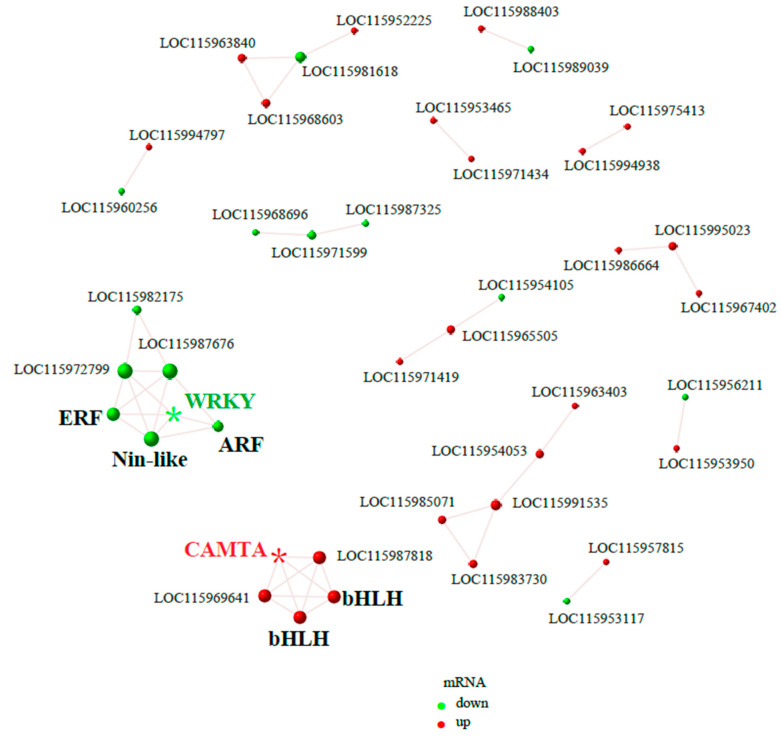
Protein interaction network diagram obtained from transcriptome data analysis. Node size is directly proportional to the number of interacting proteins. The green nodes represent proteins with significantly reduced expression levels in the transcriptome comparison database of Wanzhou acorns (high tannin content) vs. Suichang acorns (low tannin content), while the red nodes represent proteins with significantly increased expression levels. Among them, two important transcription factors WRKY and CAMTA are marked with asterisks.

**Figure 4 ijms-25-13103-f004:**
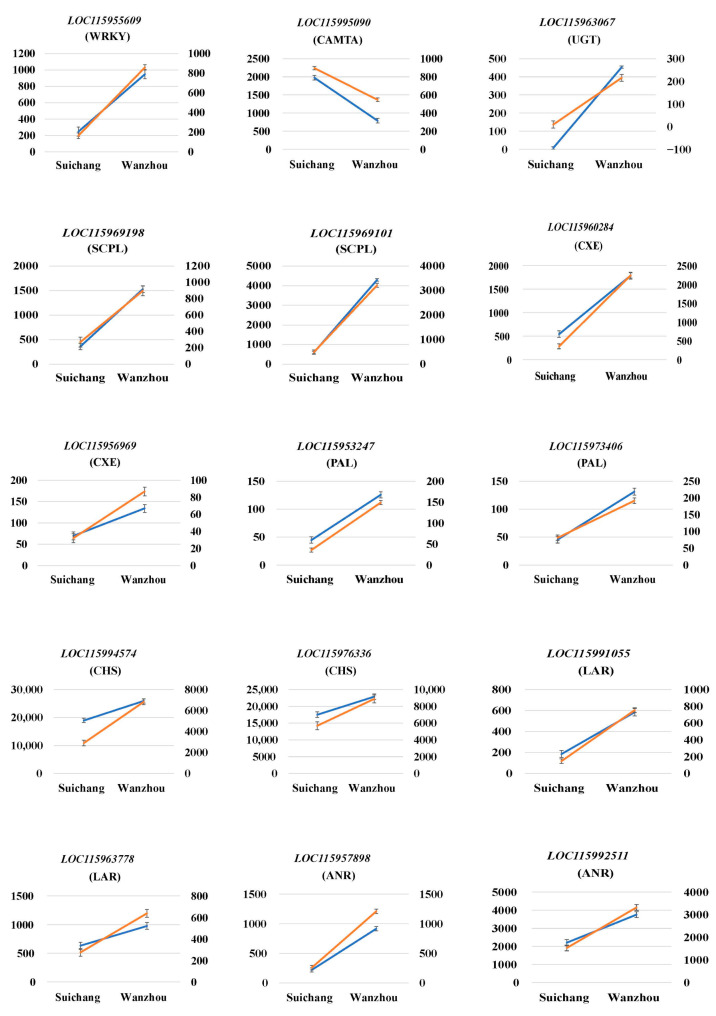
qRT-PCR validation of transcription factor gene and target gene expression levels. The left *Y*-axis represents the expression level of genes according to the FPKM value. The right *Y*-axis represents the expression level (2^−ΔΔCt^ value) of genes. The blue solid line represents the expression level of genes according to the FPKM value, the yellow solid line represents the expression level (2^−ΔΔCt^ value) of genes using qRT-PCR. The qRT-PCR values are presented as the mean ± SD of three biological replicates.

**Table 1 ijms-25-13103-t001:** Statistics of cis-acting elements in the promoters of WRKY and CAMTA transcription factor target genes.

Gene		Number of W-Box (WRKY Cis-Acting Element)	Number of CG-Box (CAMTA Cis-Acting Element)
*UGT*	*LOC115963067*		1
*SCPL*	*LOC115969198*	1	1
	*LOC115969101*	2	
*CXE*	*LOC115960284*	1	
	*LOC115956969*		3
*PAL*	*LOC115953247*	1	
	*LOC115973406*	1	2
*CHS*	*LOC115994574*	1	1
	*LOC115976336*	1	
*LAR*	*LOC115991055*	1	
	*LOC115963778*	3	1
*ANR*	*LOC115957898*	2	1
	*LOC115992511*	1	1

## Data Availability

No new data were created.

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
