# Peer review of "Transcriptomic Analysis Reveals the Opposite Regulatory Effects of WRKY and CAMTA Transcription Factors on Total Tannin Production in Quercus fabri Fruit"

_ijms, 2024, doi:10.3390/ijms252313103_

Round 1

Reviewer 1 Report

Comments and Suggestions for Authors

The article presents the interaction of two transcription factors (from the WRKY and CAMTA families) with tannin production through various enzymes. However, the article could be improved with the following points:

  1. In the introduction, there is limited discussion regarding the CAMTA transcription factor family, one of the two antagonist transcription factors identified. It would be helpful to provide more information about this family.

  2. In the protein interaction section, it might be beneficial to also analyze data using BioGRID, as STRING sometimes introduces errors, particularly in the text mining section. In the generated figure (Fig. 3), the transcription factors from each family could be marked with an asterisk or another distinguishing symbol to differentiate them from the genes they interact with. Additionally, using the protein names rather than the gene names could enhance the clarity of figure 3.

  3. The qPCR validation seems very appropriate, showing a strong correlation between both methods for all genes.

  4. A summary table with the most important genes mentioned in section 3.4 would be appreciated, so that readers do not need to refer to the supplementary data to look up specific genes.

  5. In the discussion, it would be useful to know the future directions of the research. For example, will the other transcription factors identified be further investigated? Is there a plan to create mutants for any of these genes in order to validate these results in vivo, either to study the interaction of the transcription factor with a specific region or to examine the effect of partial or total gene deletion on tannin levels?

Reviewer 2 Report

Comments and Suggestions for Authors

Dear authors,

In this study, authors re-analyzed previous transcriptome data and identified WRKY and CAMTA transcription factors (TF) to investigate their role in regulating tannin biosynthesis in fruit Q.  fabri. The study found that these TFs can comprehensively regulate the tannins biosynthesis via regulating the gene involved in this pathway, where WRKY TF exerts a positive regulatory effect on target genes, whereas the CAMTA TF imposes a negative regulatory effect.

1. The results of the study are still limited, such as not characterization of WRKY and CAMTA TFs:  the number of members of these TFs, their localization, and tissue-specific expression, etc…has not been determined/identified.

2. Functionally analysis (function verification) of the target or interest of genes in the WRKY and CAMTA transcription factor family has not been validated. For instance, authors should conduct a functional analysis of these genes (for example, via overexpression) using other plant species such as Arabidopsis,…etc. Moreover, the study’s findings would be more robust if the authors included metabolomic analysis in combination with transcriptomic analysis.

3. Fig 4. (qRT-PCR Validation of Transcription Factor Gene and Target Gene Expression Levels), the authors validated the relative expression level of genes at only one developmental stage in two regions. Authors should include these analyses (qRT-PCR and FPKM) at all three different developmental stages as which has been done for the total tannins content analysis. Thereby, it can indicate the relationship between biosynthesized tannins content and the expression level of related genes involved in the regulation of tannins biosynthesis.

4. Lacks information:  Fig 1. was presented without a y-axis (lacks title of the y-axis). It should be ”Total tannins content (unit)”. Moreover, the manuscript also lacks of section “Conclusion”. Since the section “Conclusions is not mandatory, it is better if the authors include this section.

5. Is there (based on) any reason for the author to choose Q.  fabri acorns from two regions at the three different developmental stages as: August 23, September 10, and September 23, 2021, for investigation?

6. In the sub-section “2.5 Screening of target transcription factors and determination of target genes”, the authors presented that “DNA was extracted  from  the  leaves of Q.  fabri  to serve as a  template  for further analysis.” What is the further analysis that the authors mentioned there, and what obtained data/results from these analyses? Please provide it. For this study, the author re-analyzed the previous transcriptome database and used the fruits of Q. fabri (acorns) as a target sample, not leaves.

Some Minor remarks:

- The authors should provide the GenBank/accession number of genes, which were used for qRT-PCR and listed in Table S1. In addition, forward (F) or reverse (R) primers should also be presented.

- Unclear meaning: “traditional pollution-free food”

- Double-check the caption of Fig 3.

I have marked some above comments (minor remarks) on the manuscript. Please check.

Kind regards,

Round 2

Reviewer 2 Report

Comments and Suggestions for Authors

Dear authors,

The authors have just responded to my comments by simply explaining what/how experiments were carried out and might be investigated in subsequent studies. However, some important concerns (comments 1, 2, and 3) still exist there. The authors have not provided additional data that robust the study’s findings.

The research is incomplete, and the results lack reliability. I believe this paper is not suitable for publication in the International Journal of Molecular Science.

Kind regards,

Author Response

We are very sorry for the reviewer's comments. It is difficult for us to make revisions in a short period of time because the issues raised by the reviewer require us to supplement a large number of experiments, especially the second issue. It will take several months to conduct transgenic experiments in Arabidopsis and obtain stable expression plants. Thank you for your understanding!